

# Characterization of prophages in bacterial genomes from the honey bee (*Apis mellifera)* gut microbiome

Emma K. Bueren[1], Alaina R. Weinheimer[1], Frank O. Aylward[1], Bryan B. Hsu[1], David C. Haak[2] and Lisa K. Belden[1]

[1] Department of Biological Sciences, Virginia Polytechnic Institute and State University (Virginia Tech), Blacksburg, VA, United States of America
[2] School of Plant and Environmental Sciences, Virginia Polytechnic Institute and State University (Virginia Tech), Blacksburg, VA, United States of America

Corresponding author
Emma K. Bueren, ebueren@vt.edu

## ABSTRACT

The gut of the European honey bee (*Apis mellifera*) possesses a relatively simple bacterial community, but little is known about its community of prophages (temperate bacteriophages integrated into the bacterial genome). Although prophages may eventually begin replicating and kill their bacterial hosts, they can also sometimes be beneficial for their hosts by conferring protection from other phage infections or encoding genes in metabolic pathways and for toxins. In this study, we explored prophages in 17 species of core bacteria in the honey bee gut and two honey bee pathogens. Out of the 181 genomes examined, 431 putative prophage regions were predicted. Among core gut bacteria, the number of prophages per genome ranged from zero to seven and prophage composition (the compositional percentage of each bacterial genome attributable to prophages) ranged from 0 to 7%. *Snodgrassella alvi* and *Gilliamella apicola* had the highest median prophages per genome (3.0 ± 1.46; 3.0 ± 1.59), as well as the highest prophage composition (2.58% ± 1.4; 3.0% ± 1.59). The pathogen *Paenibacillus larvae* had a higher median number of prophages (8.0 ± 5.33) and prophage composition (6.40% ± 3.08) than the pathogen *Melissococcus plutonius* or any of the core bacteria. Prophage populations were highly specific to their bacterial host species, suggesting most prophages were acquired recently relative to the divergence of these bacterial groups. Furthermore, functional annotation of the predicted genes encoded within the prophage regions indicates that some prophages in the honey bee gut encode additional benefits to their bacterial hosts, such as genes in carbohydrate metabolism. Collectively, this survey suggests that prophages within the honey bee gut may contribute to the maintenance and stability of the honey bee gut microbiome and potentially modulate specific members of the bacterial community, particularly *S. alvi* and *G. apicola*.

# INTRODUCTION

Bacteriophages, viruses that infect bacteria, are ubiquitous and the most numerous biological entities on Earth (*Hendrix, 2002*). In animals, bacteriophages are known to shape microbial communities, such as the gut or skin microbiome (*Hannigan et al.,*

*2015*). Bacteriophages are broadly classified in two forms, based on reproductive strategy. Lytic bacteriophages infect and immediately kill their bacterial hosts *via* cell lysis. In contrast, temperate bacteriophages can undergo either a lytic cycle or a lysogenic cycle, in which the phage integrates into the host genome as a prophage and replicates along with the hosts genome until its triggered to revert to a lytic lifecycle. Many phages found in animal microbial communities are lysogenic (*Minot et al., 2011*; *Kim & Bae, 2018*). Some prophages encode genes in auxiliary metabolic pathways or encode non-metabolic accessory genes ("morons"), like virulence factors, that are beneficial to their bacterial hosts and can enhance bacterial fitness (*Fortier & Sekulovic, 2013*; *Forcone et al., 2021*; *Huang et al., 2021*). For example, the human pathogens *Vibrio cholera* and *E. coli* O157:H7 can cause disease *via* toxins encoded by prophages (*Fortier & Sekulovic, 2013*).

While prophages in pathogenic bacteria have been extensively studied, the roles of prophages in other animal-associated bacteria are less clear. It is generally difficult to characterize phage-bacteria interactions in the mammalian gut due to the complexity of these microbial communities, although inroads have been made in murine models (*Hsu et al., 2019*). In contrast, the bacterial community in the gut of honey bees (*Apis mellifera*) is significantly more constrained, consisting primarily of a core group of nine bacterial phylotypes: *Bartonella apis, Bombella apis* (previously known as *Parasaccharibacter apium)*, *Frischella perrara, Snodgrassella alvi, Bifidobacterium* spp., *Bombilactobacillus* spp. (also referred to as *Lactobacillus* Firmicutes-4), *Lactobacillus* Firmicutes-5, *Gilliamella apicola*, and *Commensalibacter* spp. (also known as Alpha2.1) (*Kwong & Moran, 2016*). Some of these core microbiota, like *Gilliamella apicola, Lactobacillus* Firmicutes-5, *Bomblicatobacilus* spp., and *Bifidobacterium* spp., ferment a wide variety of carbohydrates, while others, like *F. perrara*, stimulate bee immune function (*Kwong & Moran, 2016*; *Emery, Schmidt & Engel, 2017*).

Studies using both culture and sequence-based approaches reveal the numerous roles of phages in honey bee-associated bacteria. For instance, some strains of the pathogen *Paenibacllus larvae*, which causes American Foulbrood, acquire additional virulence through the prophage encoded toxin *P1x1* (*Ebeling, Fünfhaus & Genersch, 2021*). Additionally, many phage targeting *P. larvae* have been isolated and characterized for the purpose of experimental phage therapy (*Beims et al., 2015*; *Abraham et al., 2016*; *Stamereilers et al., 2018*; *Ribeiro et al., 2019*). To capture the broader diversity of phages present in honey bee guts and include those that cannot be easily isolated, recent studies have used metagenomic surveys of viruses, in which all double-stranded DNA viruses in a sample are sequenced to characterize the "virome". These virome studies have shown that the phage community (consisting of both temperate and lytic phages) is far more diverse than the bacterial community and appears to encode auxiliary metabolic genes (*Bonilla-Rosso et al., 2020*; *Deboutte et al., 2020*; *Busby et al., 2022*). The majority of phages in these viromes were novel, and thus were not taxonomically classified. Those that were classified were primarily from the recently dissolved *Podoviridae, Myoviridae*, and *Siphoviridae* families (*Bonilla-Rosso et al., 2020*; *Deboutte et al., 2020*), leaving their current taxonomic standing unknown. However, viromes may not fully capture the phages present

in the bee gut that exist primarily as prophages because the preparation of viromes involves filtering out bacterial cells to enrich for free viral particles (*Huang et al., 2021*).

As prophages can both protect the bacterial host from infection by other phages and improve host fitness by expanded metabolic versatility, they have the potential to impact the physiology, defense, and evolution of core honey bee gut bacteria (*Deboutte et al., 2020*). To understand the roles of prophages in the regulation and function of honey bee gut bacteria, we examined publicly available honey bee gut bacterial genomes for prophage regions and compared the occurrence of prophages among bacterial species. We then assessed prophage diversity with pairwise average nucleotide identity calculations and gene-sharing networks with reference phages. To explore potential benefits the prophages confer their hosts, we functionally annotated their genes and searched for those in relevant metabolic pathways.

## MATERIAL AND METHODS

### Bacterial genomes

Publicly available genomes (complete and fragmented) of bacteria originally isolated from the gut of the honey bee, *A. mellifera*, were downloaded from GenBank between March and July, 2020 ($n = 181$) (Table 1, Table S1). The nine core bacterial phylotypes are represented in this dataset ($n = 151$ total strains across 17 species), as were two bacterial pathogens ($n = 30$). Each genome was from a uniquely named strain, and in the case of duplicate strain names in NCBI, the most complete and recent genome was used.

### Prophage identification

To detect putative prophages, we used a combination of VirSorter2 (v2.2.2; *Guo et al., 2021*, p. 2), CheckV (v.0.7.0 (*Nayfach et al., 2021*)) and VIBRANT (v1.2.1; *Kieft, Zhou & Anantharaman, 2020*). First, all genomes were analyzed with VirSorter2 (settings –include-groups "dsDNAphage,ssDNA,NCLDV,laviviridae"). Resulting viral regions were retained if they scored at least 0.5 for double stranded DNA phage (dsDNA phage, $n = 539$). Host genome regions flanking these viral sequences were then trimmed with the CheckV 'contamination' command. To exclude highly degraded phages, trimmed sequences were retained only if they were at least 5 kb in length ($n = 462$). Of these resulting sequences, a region was considered a putative prophage if it scored at least 0.9 with VirSorter2 ($n = 357$). Additionally, those with VirSorter scores of 0.5–0.9 were further analyzed with VIBRANT and were retained as putative prophage if VIBRANT also classified these sequences as virus ($n = 74$). This resulted in a final set of 431 putative prophages.

### Prophage abundance and prophage composition in honey bee bacterial isolates

Two metrics for prophage presence were assessed: the absolute abundance of prophages (total number of distinct regions) in a single bacterial genome and the portion of the bacterial genome that was composed of prophage (referred to as prophage composition). Differences in prophage abundance and composition among bacterial species were tested using Kruskal-Wallis tests. All graphs were visualized using ggplot2 and PNWColors in R (*Wickham, 2016*; *Lawlor, 2020*).

**Table 1  Honey bee-associated bacterial isolates.** The number of isolates (*n*) in each phylotype and species of the 181 publicly available genomes of unique bacterial isolates from NCBI GenBank that were downloaded March–July 2020.

|  | Phylotype | Species | Total Isolates (*n*) |
|---|---|---|---|
| Core, *n* = 150 | *Acetobacter, n = 6* | *Bombella apis* | 6 |
|  | *Bartonella apis, n = 6* | *Bartonella apis* | 6 |
|  | *Bifidobacterium spp., n = 15* | *Bifidobacterium asteroides* | 12 |
|  |  | *Bifidobacterium coryneforme* | 2 |
|  |  | *Bifidobacterium indicum* | 1 |
|  | Firmicutes-4, *n = 6* | *Lactobacillus mellifer* | 1 |
|  |  | *Lactobacillus mellis* | 5 |
|  |  | *Lactobacillus apis* | 3 |
|  |  | *Lactobacillus helsingborgensis* | 2 |
|  | Firmicutes-5, *n = 11* | *Lactobacillus kimbladii* | 1 |
|  |  | *Lactobacillus kullabergensis* | 2 |
|  |  | *Lactobacillus melliventris* | 3 |
|  | *Lactobacillus kunkeei, n = 10* | *Lactobacillus kunkeei* | 10 |
|  | *Frischella perrara, n = 2* | *Frischella perrara* | 2 |
|  | *Gilliamella spp., n = 61* | *Gilliamella apicola* | 38 |
|  |  | *Gilliamella apis* | 23 |
|  | *Snodgrassella alvi, n = 34* | *Snodgrassella alvi* | 34 |
| Pathogen, *n* = 30 | Pathogen, *n = 30* | *Melissococcus plutonius* | 17 |
|  |  | *Paenibacillus larvae* | 13 |

Larger host genomes often contain more prophage regions (*Touchon, Bernheim & Rocha, 2016*). We examined this pattern with host genome size *vs.* prophage abundance and host genome size *vs.* prophage composition using Spearman's correlation with the R library package cocor (*Diedenhofen & Musch, 2015*) and the Meng, Rosenthal and Rubin's z test (*Meng, Rosenthal & Rubin, 1992*; *Myers & Sirois*).

**Functional annotation of prophage region genes and detection of intact phages**

The 431 putative prophages were first annotated using prokka (*v.* 1.14.6) against the Prokaryotic Virus Remote Homologous Groups database (PHROGs) database (downloaded from http://millardlab.org/2021/11/21/phage-annotation-with-phrogs/ on July 20, 2022) (*Seemann, 2014*; *Terzian et al., 2021*). If an amino acid sequence had no hit to the PHROGs database (e value $< 10^{-6}$), or was classified by PHROGs as "other" or "unknown function", additional functions were then detected by aligning the amino acid sequences to hidden Markov profiles from the EggNOG 5.0 databases for bacteria, eukarya, archaea, and viruses (*Huerta-Cepas et al., 2019*) using hmmsearch (-E 0.001) with HMMER (*v.* 3.1b2) (hmmer.org, *Eddy, 2011*). Hits with bitscores above 30 were retained (*Roux et al., 2016*).

Functional composition of prophage genes were compared based on the COG category assignment within the EggNOG annotation files for bacteria, eukarya, archaea, and viruses

databases. EggNOG hits were manually re-assigned to the category of "Phage-associated" if the descriptive EggNOG name included the words: phage, baseplate, capsid, integrase, tail, tape, lysozyme, portal, holin, N-Acetylmuramoyl-L-alanine amidase, transposase, virus, or viral. Sequences that did not match to anything in the EggNOG database, or that were classified as belonging to multiple COG categories, were re-classified as COG Category S, Function Unknown. Functional composition of prophages using the combined results of PHROGs and EggNOG were then compared among bacterial host species by summing the proteins of all prophages detected in genomes of that species and visualized with the packages ggplot2 and microshades (*Wickham, 2016*; *Dahl et al., 2022*).

To estimate intactness of prophage regions, trimmed sequences were predicted as intact using the following (i) those identified as intact by PHASTER (*Arndt et al., 2016*), (ii) those that encoded at least 3 or more different phage hallmark genes (see: cornerstone genes in *Zhou et al., 2011*) and additionally encoded an integrase and/or transposase.

## Identification of unique putative prophages

To identify prophage regions likely belonging to the same population (boundary defined in *Roux et al., 2019*), genomes of the 431 putative prophages were dereplicated *via* pairwise average nucleotide identity (ANI) analysis using dRep (v3.0.0 (*Olm et al., 2017*)) (settings: –ignoreGenomeQuality -l 5000 -pa 0.95 -sa 0.95 -nc 0.85) with LASTn (v1080; *Kiełbasa et al., 2011*). Prophage pairs were considered to be from the same population if the ANI was above 95% over 85% of the shorter sequence (*Roux et al., 2019*). The representative prophage region for each population was selected with dRep. In one case, a population cluster contained two prophages that were identified from different bacterial host species. In that case, both prophage sequences were retained as representatives. As a result, a total of 237 putative prophage regions were unique to themselves and did not share high similarity with other sequences. The remaining 194 sequences had high similarity with at least one other prophage sequence, resulting in 66 distinct populations of phage. After dereplication, a total of 303 representative phage were used for downstream functional annotation and taxonomic analysis.

## Viral clustering and classification

Phages lack a universally conserved, high-resolution phylogenetic marker gene to classify them. As such, prophages are often grouped into clusters *via* gene-sharing network-based approaches or whole genome sequence alignments to reference phage (*Bolduc et al., 2017*). Dereplicated prophages in this study ($n = 303$), along with 24,289 phage reference genomes downloaded from the INfrastructure for a PHAge REference Database (INPHARED) on January 25, 2023 (*Cook et al., 2021*) were first classified with gene-sharing network approaches using vConTACT2 (v0.9.22; –rel-mode Diamond –db 'None' –pcs-mode MCL –vcs-mode ClusterON). vContact2 (v0.9.22) groups phages into subclusters, indicating phage are likely to be within the same viral genera, and broader viral clusters that indicate subfamily relatedness (relatedness somewhere between family and genus level) (*Bin Jang et al., 2019*). The resulting networks were visualized using edge-weighted spring-embedded models in Cytoscape (*Shannon et al., 2003*; *Bolduc et al., 2017*; *Bin Jang*

*et al., 2019*). Although vConTACT2 can provide taxonomic classification if target phage cluster with reference phage, for our dataset, few of the resulting clusters included references phages that enabled classification at any taxonomic level. Therefore, while vConTACT2 gene-sharing networks were used to assess the relationships among honey bee prophages, we ended up relying more on whole genome alignments to reference phage to assign broader taxonomy.

To assign taxonomy based on similarities to reference viral groups, we compared the average amino acid identity (AAI) of proteins between our dereplicated putative prophages and phage from a dereplicated *Caudoviricetes* database (amino acids downloaded from the INPHARED on November 5, 2022, taxonomic metadata downloaded from INPHARED on March 16, 2023) (*Cook et al., 2021*). The 16,253 reference genomes were first dereplicated with dRep (–ignoreGenomeQuality -p 32 -l 20000 -pa 0.90 –SkipSecondary inphared_derepout_90mash) resulting in 2,600 representative reference genomes. Amino acid sequences of proteins encoded by the reference were predicted using prodigal (default on each genome). A BLAST database was made of the reference proteins (makeblastdb). The honey bee gut prophages protein predictions (produced *via* prodigal, described above) were aligned to this reference database with BLASTp (*E* value 0.001). Taxonomic class was assigned at the most precise rank possible using each prophage's AAI similarity to its top hitting reference phage. If a putative prophage and a reference phage shared greater than 70% AAI across 85% of proteins, it was classified as belonging to the same genus as the reference (*Turner, Kropinski & Adriaenssens, 2021*). If a phage did not meet this threshold but still shared at least 30% AAI over 50% of proteins, it was classified as the same family as the reference phage (*Turner, Kropinski & Adriaenssens, 2021*). If a prophage could not be classified using the above thresholds, but its top reference hit was to a *Caudoviricetes* class phage, it was classified broadly as *Caudoviricetes*. Any putative prophage with a top reference hit to a completely unclassified phage was labeled as Unclassified.

# RESULTS

## Distribution of prophages across core bacterial hosts

Bacterial host species significantly varied in both the number and composition of prophages across all bacterial isolates, both core and pathogens (Kruskal-Wallis, $p < 0.0001$; Fig. 1 & Table S2). A total of 269 high-confidence prophage regions were predicted across the 151 core bacterial isolates before dereplication. Bacterial genomes ranged in size from 1.3 Mb–3.5 Mb (Table S2). Among the core bacterial species, the number of prophages per isolate varied from 0 to 7 and prophage composition of the bacterial genome varied from 0–7.01% (Table S2).

The core bacteria with the highest median number of prophages and prophage composition was *S. alvi* (3.0 ± 1.46 prophages; 2.58% ± 1.40 prophage composition) and *G. apicola* (3.0 ± 1.58 prophages; 2.04% ± 1.48 prophage composition) (Figs. 1A and 1B, Table S2). Notably, while *Bombella apis, Bartonella apis*, and *F. perrara* had the second highest median number of prophages per genome, *L. melliventris* had the second highest

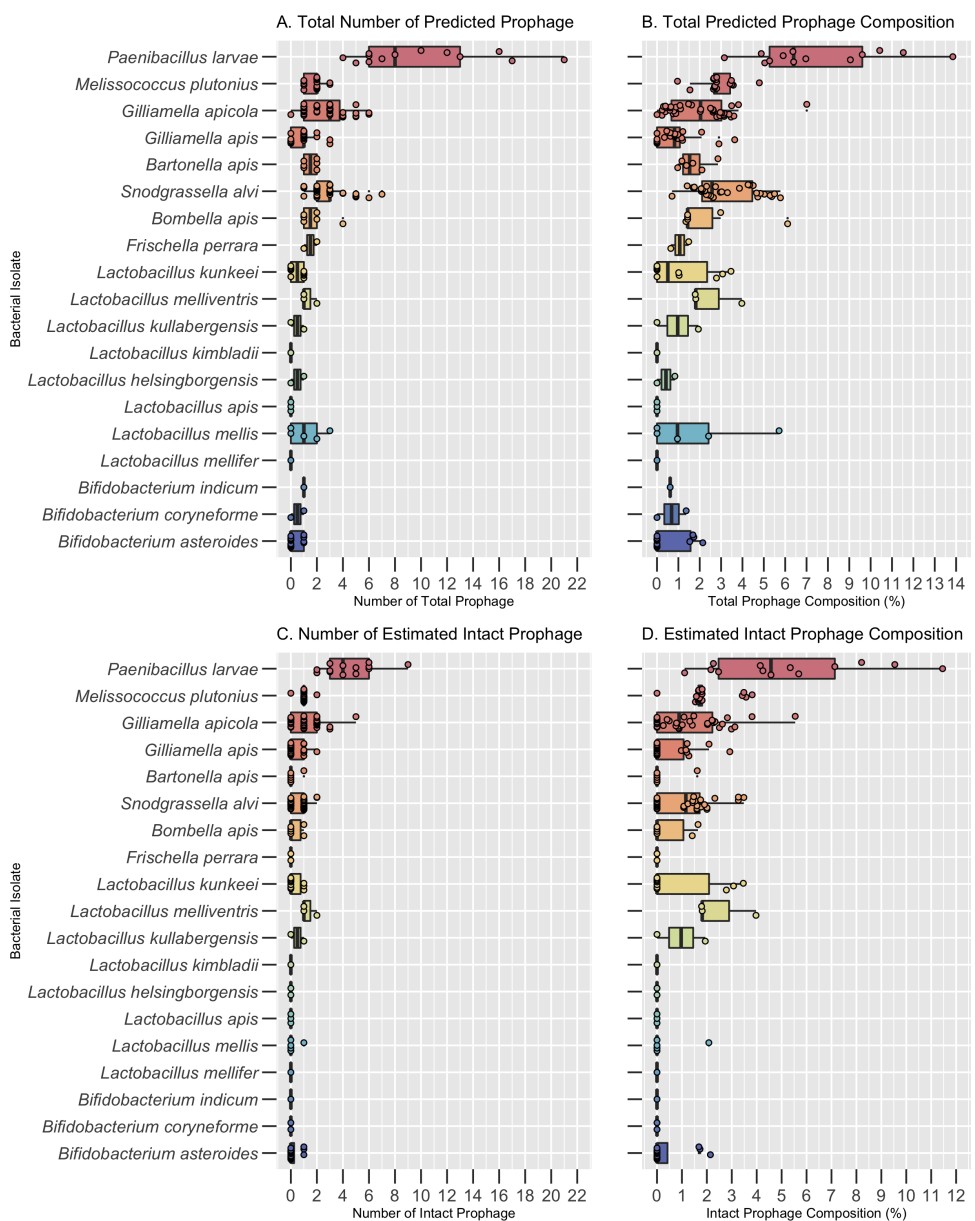

**Figure 1 Prophage frequency and composition across bacterial hosts.** (A) The predicted number of to-tal prophage regions, (B) total prophage composition (the percent of the bacterial genome composed of prophages), (C) number of intact prophage regions, and (D) intact prophage composition per bacterial isolate. Dots represent individual bacterial isolates, while the center line indicates the median value for each bacterial species. The boxes span the interquartile range and whiskers represent the 25th and 75th quartile ranges. Values below or above the 25th and 75th quartiles are indicated by the smaller circles next to individual dots. The minimum, maximum, and median values are listed in Table S2.

prophage composition (Figs. 1A and 1B; Table S2). All other core species had a median number of 1 prophage or less and a prophage composition of less than 1% (Figs 1A and 1B, Table S2). These included *Bifidobacterium asteroides, Lactobacillus mellifer, Lactobacillus*

*apis*, and *Lactobacillus kimbladii*, which had a median of 0 prophages per genome (Figs 1A and 1B; Table S2).

Of these 269 putative prophage regions, 90 were estimated to be intact. Bacterial hosts with the highest median number of estimated intact prophages were *G. apicola* (1.0 ± 1.1), *S. alvi* (1.0 ± 0.60) and *L. melliventris* (1 ± 0.58) (Fig. 1C; Table S2). All three isolates of *L. melliventris*, 71% of *G. apicola* isolates ($n = 38$) and 56% of *S. alvi* isolates ($n = 34$) possessed at least one estimated intact prophage (Table S3). The second highest median of intact prophage was found in *L. kullabergensis* (0.5 ± 0.71), with one of the two *L. kullabergensis* isolates possessing an intact prophage (Fig. 1C; Table S3). All other core species had a median of 0 intact prophages (Fig. 1C, Table S2). However, intact prophages did still occur in these isolates at lower frequencies (Table S3). Intact prophage composition varied slightly compared to prophage abundance, with *L. melliventris* having the highest composition (1.81% ± 1.25), followed by *S. alvi* (1.16% ± 1.09), *L. kullabergensis* (0.97% ± 1.37), and *G. apicola* (0.89% ± 1.31) (Fig. 1D; Table S2).

## Distribution of prophages across pathogenic bacterial hosts

A total of 162 prophage regions were predicted across the 30 bacterial isolates from the two pathogens, *P. larvae* and *M. plutonius*, before dereplication. The bacterial isolate genomes ranged in size from 2.0 Mb–4.8 Mb (Table S2). The number of predicted prophage regions ranged from 1 to 22, and prophage composition ranged from 0.96%–13.85% of the bacterial genome (Table S2). *P. larvae* had the highest median number of prophages per genome (8.0 ± 5.33) and prophage composition (6.40% ± 3.08) across all bacterial species, including the core bacteria (Figs. 1A and 1B; Table S2). The median number of prophages found per genome of *M. plutonius* was 2.0 ± 0.64, and its median prophage composition was 2.77% ± 0.83 (Figs. 1A and 1B; Table S2).

Of the 13 *P. larvae* isolates analyzed, all had at least one intact prophage, with a median of 4 ± 1.98 intact prophage per isolate (TableS-Freq; Fig. 1C; Table S2). *M. plutonius* had a median of 1 ± 0.35 intact prophage, with 16 of 17 isolates possessing an intact prophage (Fig. 1C; Table S2; TableS-Freq). The total intact prophage composition of *P. larvae* (4.57% ± 3.10) was higher than *M. plutonius* (1.71% ± 0.97) (Fig. 1D; Table S2).

## Prophage abundance is more correlated with bacterial genome size than prophage composition

Bacterial genome size was positively correlated with both the number of prophages (Spearman's $\rho = 0.55$, $p < = 1.6e{-}15$, Fig. 2A) and prophage composition (Spearman's $\rho = 0.34$, $p = 2e{-}6$, Fig. 2B). Meng, Rosenthal and Rubin's correlation comparison determined that bacterial genome size had a stronger correlation with the number of prophages than with prophage composition ($z = 5.9819$, $p$-value $< 0.001$, 95% CI [0.18–0.35]) (*Meng, Rosenthal & Rubin, 1992*).

## Prophage-encoded genes varied by bacterial host

Nearly half (48.4%) of all predicted protein-encoding genes of the dereplicated prophages were classified broadly as phage-associated. A total of 2.3% of all genes were classified by PHROGs specifically as "Moron, auxiliary metabolic genes, and host takeover" (Table

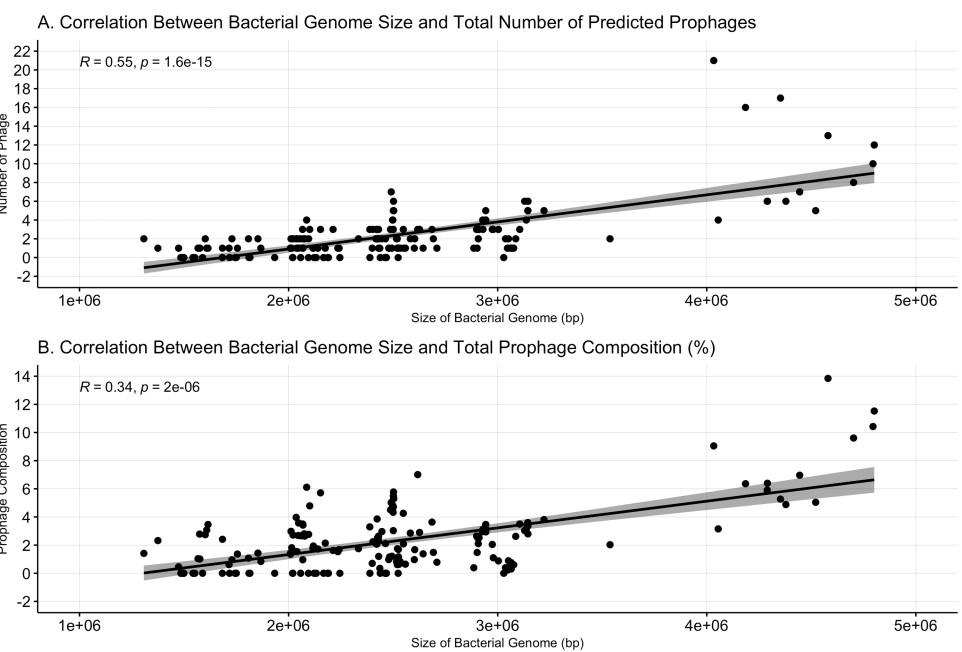

**Figure 2 Bacterial genome size correlates to prophage number and composition.** (A) The number of prophages identified and the (B) percent prophage composition is positively associated with bacterial genome size. Dots represent individual bacterial genomes.

S5). *S. alvi* had the highest frequency of genes in this category (66 genes, 3.3% frequency), due to its large collection of phage toxins and antitoxins (Fig. 3). These toxins shared homology to Doc-like toxins (10 genes), BstA abortive infection systems (five genes), HicB-like toxin-antitoxin systems (17 genes), and a RelE-like toxin (one gene), among other undescribed toxins (33 genes) (Table S7). *P. larvae* had the second highest number of genes (124 genes, 2.9% frequency), in the PHROGs category: moron, auxiliary metabolic genes, and host takeover (Fig. 3). *P. larvae* prophages possessed a variety of phage toxins, antitoxins, and additionally, genes associated with bacteriocin production or immunity (Table S7). Some *P. larvae* prophages also possessed a gene associated with quaternary ammonium compound-resistance proteins (two genes), or to ABC transporters (seven genes), indicating the presence of antimicrobial resistance genes (Table S7). One other antimicrobial-associated gene in a *P. larvae* prophage, originally classified as "unknown" by PHROGS, was annotated as a bacitracin resistance protein, BacA, by the eggNOG database (Table S7). Additionally, *P. larvae* prophages possessed genes for flavodoxin, a NifU-like Fe-S cluster assembly protein, and a phosphoadenosine phosphosulfate (PAPs) reductase (Table S7). The PAPs reductase also occurred frequently in prophages isolates from *G. apicola* (13 genes), and once in a *B. asteroides* prophage (Table S7).

The majority of remaining predicted protein-encoding genes (49.0%) were classified broadly as Function Unknown (COG Category S or no hits to the COG database) after failing to match to the PHROGs database (Table S5). Coding regions that were able to be classified broadly by COG were distributed into the categories of Information
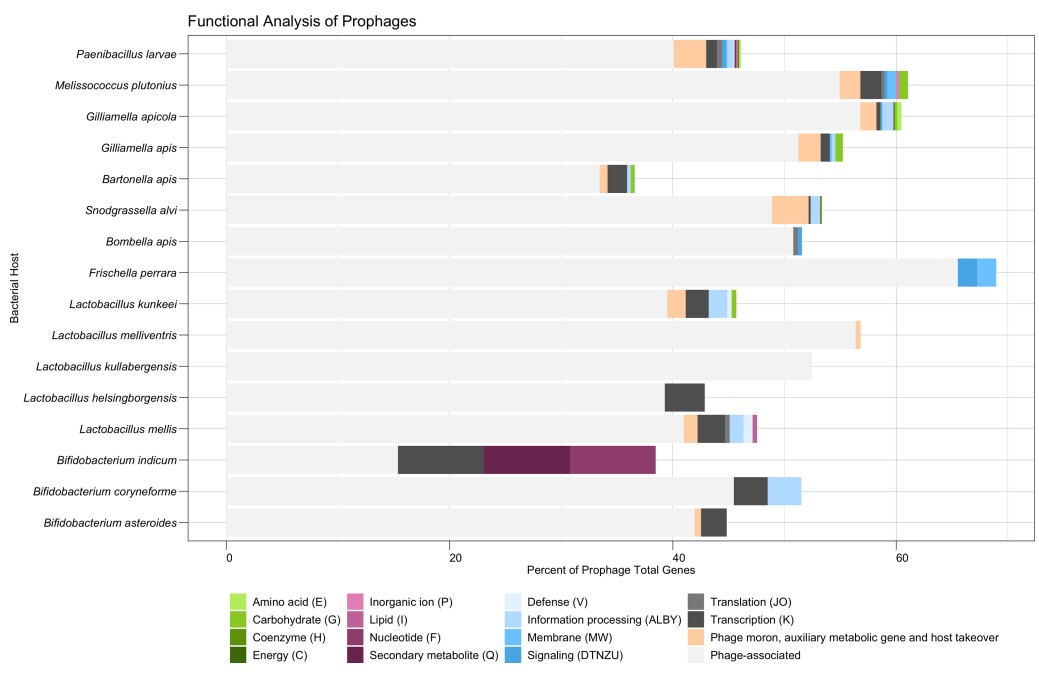

**Figure 3** **Functional analysis of prophages found in honey bee-associated bacteria.** The distribution of phage-associated and COG categories assigned to prophage genes, organized by bacterial host. Similar COG categories are grouped together for visual clarity. *X*-axis indicates percent. COG Category S (Unknown or No Hits) is not shown.

Storage and Processing (1.6%), Metabolism (0.5%), or Cellular Processing and Signaling (0.5%) (Table S4). Within these broader categories, COG category K (Transcription—Information Storage and Processing) occurred most frequently at 0.8% (Table S5). Amino Acid Transport and Metabolism (COG category E) and Carbohydrate Transport and Metabolism (COG category G) were the most commonly occurring metabolic categories, each corresponding to 0.1% of genes (Table S5, Table S7). Genes associated with amino acid metabolism were found in prophages of *G. apicola* (eight genes, 0.4% frequency) and *P. larvae* (five genes, 0.1% frequency) (Fig. 3, Table S6). Five different prophages of *P. larvae* appear to possess a gene homologous for a thermolysin metallopeptidase, while two share a gene associated with glycine amidinotransferase activity (Table S7). The amino acid genes associated with *G. apicola* prophages are more varied but appear to be associated with arginosuccinate and glutamate (Table S7). Prophage genes associated with carbohydrate metabolism were found in several bacterial hosts: *M. plutonius* (three genes, 0.8% frequency), *G. apis* (four genes, 0.7%), *L. kunkeei* (one gene, 0.41% frequency), *Bartonella apis* (one gene, 0.3% frequency), *G. apicola* (four genes, 0.2% frequency), and *P. larvae* (two genes, <0.1% frequency) (Fig. 3, Table S6). Interestingly, a single *M. plutonius* prophage possessed all three carbohydrate-associated proteins identified for that bacterial host: a mannitol dehydrogenase, a MFS/sugar transport protein, and a protein associated with the dehydration of D-mannotate (Table S7). Similarly, a single *G. apicola* prophage possessed all four of the carbohydrate-associated proteins identified for that bacterial

host species: three proteins associated with the glycerate kinase family and one associated with alpha-ribazole phosphatase activity (Table S7). In contrast, three separate prophages of *G. apis* possessed a gene for alpha-galactosidase, with one also carrying a gene for an exopolysaccharide biosynthesis protein (Table S7). Additionally, two separate prophage species of *P. larvae* both possessed a gene associated with the D-gluconate metabolic process (Table S7). The remaining COG metabolic categories appear to be more rare, although they did still occur (Fig. 3, Table S6). For example, the sole *B. indicum* prophage that was identified possessed genes associated with nucleotide metabolism (a purine-cytosine permease) and secondary metabolite metabolism (isochorismatase) (Table S7).

## Closely related prophages typically share host species

Using vConTACT2, a total of 208 honey bee prophages from the dereplicated prophages ($n = 303$) were grouped into 56 subclusters, with the remaining prophages detected as outliers ($n = 82$) or unable to be incorporated into the network ($n = 6$) (Fig. 4). The bacterial host strongly predicted clustering, with prophages of the same host species typically forming exclusive subfamily clusters (Fig. 4A). Only 10 of the 62 predicted subclusters included a prophage from more than one host, *via* either multiple honey bee bacterial species or the inclusion of non-honey bee reference phage (Fig. 4A). In several of these mixed clusters, the bacterial host genus was shared among isolates even if the species was not (Table S8).

In the resulting gene sharing network, honey bee prophages within the same clusters, and therefore often from the same bacterial host genome, were most closely positioned to each other (Fig. 4B). However, prophages from different host species sometimes appeared to share genetic similarities. For example, prophages from *G. apicola* and *G. apis* were closely positioned, as were some of the prophages from *S. alvi* and *Bombella apis*, and *Bartonella apis* (Fig. 4B). Interestingly, most of the prophages from *Latobacillus* spp., appear to share genes with each other, but prophages of *L. kunkeei* appear very diverse potentially due to few genes shared among these prophages (Fig. 4B). Furthermore, *L. mellis* prophages appear to share some similarity to prophages isolated from *M. plutonius* and *P. larvae* (Fig. 4B).

## Most prophages from honey bee symbionts are unclassified *Caudoviricetes*

Of the 303 distinct prophage populations, 59.4% (180 prophage regions) were only able to be classified as *Caudoviricetes*, while an additional 31.7% prophages (96 prophage) were unable to be taxonomically classified using top reference hits (Fig. 5). Only the remaining 27 prophage regions (0.9%) were able to be classified at a family level or lower (Fig. 5). A total of 12 (16%) *G. apicola* prophages ($n = 75$) belonged to the family *Peduoviridae (Class: Caudoviricetes)*, while one of three *F. perrara* prophages was predicted to belong to the *Mesyanzhinovviridae* family (Class: *Caudoviricetes*) (Fig. 5). The most common taxonomic ranks assigned to prophage from *P. larvae* isolates ($n = 83$) outside of *Caudoviricetes* (69 prophage, 66.3%) or unclassified (21 prophage, 20.2%) were the genera *Fernvirus* (seven prophage, 6.7%), *Vegasvirus* (three prophage, 2.9%) and *Halyconevirus* (two prophage,
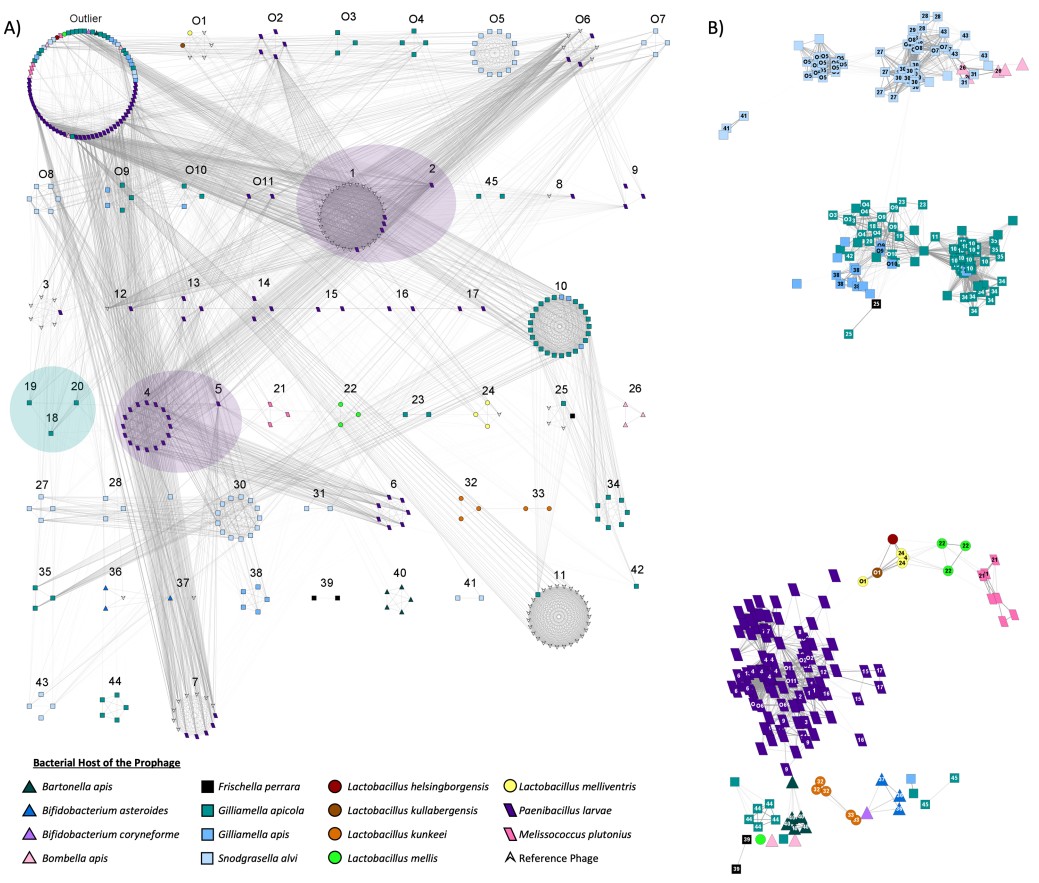

**Figure 4** **Viral clustering and network analysis based on shared genes.** Nodes represent individual prophages. Bacterial hosts are indicated by the node's color and shape. (A) Prophages are separated based on sub-family viral clusters. Numbers above clusters indicate a specific subcluster. Distance between subclusters has no significance, but viral subclusters that come from the same greater viral cluster (indicated by transparent oval overlay) are more related to one another than phage originating from other clusters. Subclusters O6, 1, 7, 8, and 12 contain reference *P. larvae* phage from external studies. Prophage within the outlier cluster are not related to one another. (B) The layout of the network is based on shared genes, with strength indicated by opacity of the network edges. The numbers on nodes indicate subcluster. Blank nodes indicate outliers, while subclusters starting with "O" indicate unique overlapping subclusters.

1.9%) (Class: *Caudoviricetes*) (Fig. 5). A single *P. larvae* prophage each (1%) was classified as either the genera *Lilyvirus* or *Dragolirvirus* (Class: *Caudovircetes*).

## DISCUSSION

By analyzing publicly available genomes, 303 distinct putative prophage regions were predicted from honey bee-associated bacteria. Although these predicted prophage regions have not been empirically confirmed to be intact and active, this is the first targeted survey of dsDNA prophage distribution among core honey bee bacterial symbionts. While some of these predicted prophage regions may be genomic relics of prophages, the minimum threshold of 5 kb ensures that all identified prophages, active or relic, could possess genes that may affect their bacterial hosts. Additionally, it should be noted that this study

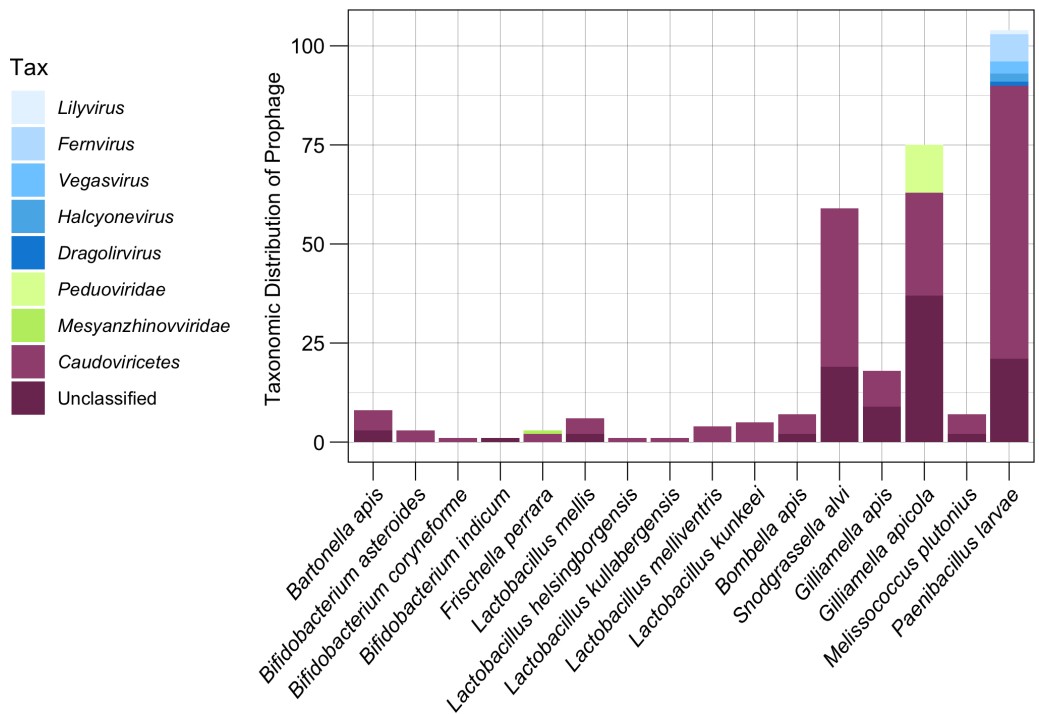

**Figure 5 Taxonomic classification of prophages.** The lowest taxonomic rank (genus, family, or class) assigned to each prophage based on amino acid similarity to its closest reference phage. Genus and family classifications were assigned if prophages shared AAI across all proteins with a reference above respective thresholds. Prophage which did not meet these thresholds but still matched to a *Caudoviricetes* reference were broadly classified as *Caudoviricetes* class. Prophage which matched to unclassified reference phage were grouped as Unclassified.

targeted specifically dsDNA phage. The prevalence of prophages with single stranded DNA genomes, like those found in the families *Microviridae* or *Inoviridae*, have yet to be determined (*Székely & Breitbart, 2016*).

The wide presence of distinct prophage regions found in this study supports the broader trend of high viral diversity in the honey bee gut. However, some of the most common hosts of prophage identified in this study differ from the predicted hosts of viral particles revealed in metagenomic studies of the gut virome (*Bonilla-Rosso et al., 2020*; *Deboutte et al., 2020*; *Busby et al., 2022*). This is possibly because viral metagenomes filter out bacterial cells to enrich for free viral particles. The differences in the host ranges identified with metagenomic approaches and our study of individual isolate genomes may be partially due to the phenomenon of superinfection exclusion, in which bacteria that contain lysogens are protected from additional infection of phages (*Bondy-Denomy et al., 2016*). For instance, we found that *S. alvi* contains a median of three prophage regions, and about half of all *S. alvi* isolates possessed at least one intact prophage. However, *S. alvi* phages have been relatively rare in metagenomic virome analyses of the honey bee gut (*Bonilla-Rosso et al., 2020*; *Busby et al., 2022*). It is possible that bacterial hosts like *S. alvi* may be underrepresented in metagenomic studies due to the challenge of predicting hosts
of viral contigs using CRISPR spacers (*Dion et al., 2021*). However, *S. alvi* prophages may also limit the population of lytic phages targeting *S. alvi,* or intact prophages in *S. alvi* may excise infrequently; these factors could make detection of *S. alvi* prophage in metagenomic studies less likely. Furthermore, in one virome study, 25% of the phage identified in the honey bee gut virome were predicted to infect *Bifidobacterium* spp. (*Bonilla-Rosso et al., 2020*), but few *B. asteroides* and other *Bifidobacterium* spp. genomes analyzed in our study had estimated intact prophage; only three out of 12 *B. asterioides* isolates possessed an intact prophage, while the single *B. indicum* isolate and the two *B. coryneforme* isolates did not possess any. This may suggest that some honey bee symbionts, like *Bifidobacterium* spp., are targeted predominantly by lytic phages, or alternatively, highly active temperate phages that frequently excise from their host.

In contrast, *G. apicola* has a high number of prophage regions per genome, which is consistent with other virome reports that assessed the abundance of phage (both lytic and temperate) that existed outside of the bacterial cell at the time of sampling (*Bonilla-Rosso et al., 2020*; *Busby et al., 2022*). In this study, *G. apicola* genomes had a higher frequency of intact prophage than many other core bacteria. It is possible that unlike *S. alvi*, these prophages may frequently release progeny phage into the host gut, contributing to the overall composition of the virome. Alternatively, the prophage of *G. apicola* may fail to provide significant superinfection exclusion and protection from lytic phage. A similar scenario may be true for *L. melliventris*, which had a predicted intact prophage in every isolate and was also predominant in one of the same virome studies (*Busby et al., 2022*). Investigating the lifestyle of phage identified in metagenomic studies may further elucidate if certain bacterial hosts are primarily targeted by a specific lifestyle of phage. In at least one study, the majority of *Bifodibacterium* spp. were targeted by phage predicted to be lytic, while *Gilliamella* spp. phage were primarily predicted to be temperate (*Bonilla-Rosso et al., 2020*). Exploration of additional bacterial metagenomic studies would also further clarify the relationship between certain phylotypes of bacteria and prophage, as our study is limited by having few sequenced isolates of several of the bacterial species. For example, conclusions about *L. melliventris* should be interpreted cautiously, as only three genomes were publicly available at the time of our analysis. Furthermore, additional genomes from species less represented in this study, such as *F. perrara*, many of the other *Latcobacillus* spp., and *Bifidobacterium* spp., would clarify which bacteria harbor prophages more frequently and to what degree these prophages remain excisable. It should also be noted that the bacterial species with more sequenced isolates, such as *Gillamellia sp.*, and *S. alvi*, typically had a higher median abundance of prophages, possibly indicating a bias in this analysis.

The prevalence of prophages across bacterial species is also associated with bacterial traits, such as pathogenicity (*Knowles et al., 2016*; *Silveira & Rohwer, 2016*; *Touchon, Bernheim & Rocha, 2016*). The two bacterial pathogens in our study, *P. larvae* (American Foulbrood) and *M. plutonius* (European Foulbrood), noticeably differ from each other in terms of both prophage number and prophage composition. *P. larvae* has a higher number of prophages and prophage composition compared to both *M. plutonius* and all core bacterial species. In contrast, the number of prophages and prophage composition of *M. plutonius* is closer to the core species *S. alvi* and *L. melliventris*. While the prophage-encoded toxin *P1x1* found

in some strains of *P. larvae* is known to contribute to virulence, it is not the sole cause of virulence; genotypes lacking *P1x1* can still cause severe disease (*Ebeling, Fünfhaus & Genersch, 2021*). However, the high abundance and diversity of putative prophage regions in nearly all *P. larvae* isolates may imply that there are more prophage-encoded virulence factors to be discovered. For example, the presence of *P. larvae* prophage regions containing possible antimicrobial resistance genes, as well as carbohydrate, iron and sulfur metabolism genes, may indicate additional ways phage provide fitness to the pathogen (discussed in *Ribeiro et al., 2022*). The presence of three separate carbohydrate-associated genes in a single *M. plutonius* prophage indicates that prophage-encoded auxiliary metabolic pathways provide additional benefit to *M. plutonius*, as well. This is particularly interesting because *M. plutonius* strains are not typically able to metabolize the sugar mannitol, but can metabolize D-mannose (*Arai et al., 2012*). The phage-encoded mannitol dehydrogenase may break down mannitol into D-mannose, while a second phage-encoded protein may then dehydrate the D-mannose to release energy. The third phage-encoded protein, a sugar transporter, could have a role in shuttling these sugars in or out of the bacterial cell. Combined, these three prophage-encoded genes may serve to enhance the pathogenicity of *M. plutonius* 82 by providing an additional sugar to exploit. However, the lower prophage abundance and composition in *M. plutonius* may indicate that prophages play a less significant role in the virulence of *M. plutonius* compared to *P. larvae*.

Functional annotation of the predicted prophages also suggests that some commensal bacterial, such as *Gilliamellia* spp., may receive auxiliary metabolic benefits from their prophages. *Gilliamella* spp. contribute to the breakdown of pollen by primarily targeting pectin (*Engel, Martinson & Moran, 2012*). It is possible that the glycerate kinases and the alpha-ribazole phosphatase found in *G. apicola* prophage, or the alpha-galactosidase found in three separate *G. apis* prophages assist with the breakdown of pectin or the production of other metabolites. Some isoforms of alpha-galactosidase have been shown to be associated with pectin hydrolysis in fruits (*Soh, Ali & Lazan, 2006*). The predicted glycerate kinase may be used to support bacterial glycolysis pathways, while the alpha-ribazole phosphatase enzyme may contribute to cobalamin biosynthesis, an important secondary metabolite (*Doughty, Hayashi & Guenther, 1966*; *O'Toole, Trzebiatowski & Escalante-Semerena, 1994*). Additionally, an exopolysaccharide biosynthesis coding region found in a *G. apis* prophage may contribute to biofilm formation. Biofilms produced by *Gilliamellia* spp. may inhibit pathogen invasion in the midgut (*Engel, Martinson & Moran, 2012*). As a result, it is possible that prophage presence in *Gilliamella* spp. may directly benefit its bacterial host, and consequently the honey bee. Additionally, several *G. apicola* prophages appear to carry a PAPS reductase gene, which may enhance the host's ability to metabolize sulfur (*Mara et al., 2020*).

Given that phage typically have a high host-specificity, along with the ability to recombine with host DNA and other co-infecting phages, it is not surprising that the prophage regions found in the same host species are most closely related to each other (*Casjens, 2003*; *Brüssow, Canchaya & Hardt, 2004*). At the same time, phages from entirely different bacterial host species may occasionally transfer genes horizontally to each other (*Stecher, Maier & Hardt, 2013*), and this may be more likely to occur between phages with hosts that are physically

near each other. Prophages identified from bacterial hosts in the midgut and ileum (*Bombella apis, Bartonella apis, Gilliamellia* spp., *F. perrara*, and *S. alvi*) were positioned more closely in the network, and therefore more likely to share genes with each other, than prophages of bacteria in the rectum (*Bifidobacterium* spp., *Lactobacillus* spp.), possibly because the phages are either more closely related to one another or due to increased horizontal gene transfer between nearby phage communities (*Anderson & Ricigliano, 2017*). The idea of spatially-separated phage communities in the gut is supported by a recent study investigating the gut virome of the marine invertebrate *Ciona intestinalis*, which found phage communities were localized to specific regions (*Leigh et al., 2018*). Due to the inherent difficulty in classifying phages, and the limited number of International Committee on Taxonomy of Viruses (ICTV) classified reference phage genomes relative to the vast diversity of bacteriophages in nature, it is unsurprising that a large portion of predicted prophage regions in our study remain unclassified (*Bin Jang et al., 2019*). Less than ten percent of the predicted prophages were classifiable at a family level or lower. A few prophage predicted from *P. larvae* shared taxonomic similarity to reference *P. larvae* phage isolated from other studies, but the majority were only classifiable as belonging to *Caudoviricetes* order, indicating that a greater diversity of *P. larvae* phage may exist (*Stamereilers et al., 2018*).

Although honey bee gut bacterial communities contain relatively few bacterial phylotypes, prophages may contribute to the high bacterial strain diversity within the honey bee gut microbiome (*Ellegaard et al., 2015*; *Ellegaard & Engel, 2019*), as 237 out of the 303 unique prophage regions were detected in only one bacterial isolate. This may indicate that in some bacterial species, susceptibility to specific prophage integration may vary depending on host strain genotype, possibly driving both viral and bacterial evolution or diversification. By potentially impacting their host's evolution, active honey bee gut prophages may affect general honey bee health and dysbiosis, particularly within the context of antibiotic and pesticide exposure. In other host-associated and environmental systems, antibiotics and other chemicals can induce prophage excision (*Danovaro et al., 2003*; *Allen et al., 2011*). Frequent exposure to either antibiotics or pesticides, something that honey bees must contend with (*Kakumanu et al., 2016*; *Raymann, Shaffer & Moran, 2017*), may result in a gut state that is constantly perturbed, leading to prophage induction and further gut perturbation. On the other hand, prophages in the honey bee gut may stabilize bacterial communities against antimicrobials and pesticides, as prophages can also enhance the fitness of their bacterial partners *via* beneficial accessory genes that provide resistance to antibiotics or other environmental pollutants (*Wang et al., 2010*; *Huang et al., 2021*).

The diverse prophage community identified in our study supports the well-established idea that prophages are common and present in a wide range of bacterial hosts (*Touchon, Bernheim & Rocha, 2016*). However, some of the most common bacterial hosts of the prophage community identified in this study do not entirely reflect that of the wider virome studies of honey bee guts, indicating that prophages may remain cryptic despite more targeted viral sequencing approaches. As a result, the role prophages play in bacterial

evolution and community dynamics will require further untangling both within the honey bee system and beyond.

## CONCLUSION

As the first step in identifying how the prophage community interacts with and modulates bacterial communities of the honey bee gut, a survey of prophage in publicly available honey bee gut-associated bacterial genomes was conducted. This study found that prophage abundance and composition vary across the core honey bee gut bacterial species, with *S. alvi* and *G. apicola* possessing the highest prophage number and percent phage composition, and *L. melliventris* having the highest frequency of likely intact prophage. Interestingly, there was some discrepancy between the most commonly predicted bacterial hosts for prophages compared to phage isolated from metagenomic studies. This may indicate that certain nuances of prophage-bacteria interactions are not able to be captured by viromes, which primarily focus on free viral particles. Additionally, the prophages of some commensal bacterial hosts, such as *Gilliamella* spp., may provide auxiliary metabolic genes for carbohydrate metabolism, which could possibly benefit the honey bee host. These results set the foundation for targeted exploration of the prophage infecting core bacterial species through culture-based methods.

### Funding

This research was supported by the National Science Foundation (MCB-1817736). The funders had no role in study design, data collection and analysis, decision to publish, or preparation of the manuscript.

### Grant Disclosures

The following grant information was disclosed by the authors:
National Science Foundation: MCB-1817736.

### Competing Interests

The authors declare there are no competing interests.

### Author Contributions

- Emma K. Bueren conceived and designed the experiments, performed the experiments, analyzed the data, prepared figures and/or tables, authored or reviewed drafts of the article, and approved the final draft.
- Alaina R. Weinheimer conceived and designed the experiments, performed the experiments, authored or reviewed drafts of the article, and approved the final draft.
- Frank O. Aylward conceived and designed the experiments, authored or reviewed drafts of the article, and approved the final draft.
- Bryan B. Hsu conceived and designed the experiments, authored or reviewed drafts of the article, and approved the final draft.

- David C. Haak conceived and designed the experiments, authored or reviewed drafts of the article, and approved the final draft.
- Lisa K. Belden conceived and designed the experiments, authored or reviewed drafts of the article, and approved the final draft.

## Data Availability

The sequences are available at Virginia Tech's Data Repository: Bueren, Emma (2023): HBProphage PeerJ 2023. University Libraries, Virginia Tech. Dataset. https://doi.org/10.7294/22203001.v1.

The code and raw data are available at GitHub and Zenodo: https://github.com/ebueren/HBProphage_PeerJ.

ebueren. (2023). ebueren/HBProphage_PeerJ: 042623-HBProphagePeerJRelease (PublicationDOI). Zenodo. https://doi.org/10.5281/zenodo.7869127.

## Supplemental Information

Supplemental information for this article can be found online at http://dx.doi.org/10.7717/peerj.15383#supplemental-information.

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
