# Peer review of "Characterization of prophages in bacterial genomes from the honey bee (Apis mellifera) gut microbiome"

_PeerJ, doi:10.7717/peerj.15383_

## Round 0.1 · original submission · Minor Revisions

All reviewers of your manuscript found it very interesting, well-written, and worthy of publication. But there are a number of suggestions provided by the reviewers that need to be addressed before the manuscript can be accepted for publication. Please provide a detailed response to each item raised in the reviewer critiques. This should include revisiting the taxonomic classification of the phage, especially in consideration of the current, extensively revised taxonomy of the Class Caudoviricetes by the ICTV.

Reviewer 1 ·

Basic reporting

Editors,

The manuscript entitled, “Characterization of prophages in bacterial genomes from the honey bee (Apis mellifera) gut microbiome” by Bueren et. al describes their analyses of database accessible sequence information to examine “prophages in 17 species of core bacteria in the honey bee gut and two honey bee pathogens’, which is an important step in understanding the potential impact of bacteriophage on their bacterial hosts, as well as potentially the host organisms for the bacteria (i.e. honey bees). The manuscript, including the methods section, was written clearly. The authors did a nice job citing previous and they did not over interpret their data, nor over speculate from their findings. The detail included in the methods section ensures that others could reproduce and/or re-evaluate, with the inclusion of additional sequence data, these findings in future studies. My only minor/small suggestion is that in Figures 1 & 5, the authors should consider italicizing the bacterial names.

Experimental design

see above

Validity of the findings

see above

Reviewer 2 ·

Basic reporting

In this study, Bueren et al. explore the prophage content of 181 bacterial genomes isolated from the honey bee gut. They identify a total of 303 high-confidence putative prophages. The authors show a high variability in prophages total number and prophage composition between different species of honey bee gut-associated bacteria. On one hand, the honeybee pathogens P. larvae and M. plutonius have the highest prophage number and composition. On the other hand, the core bacteria of the honeybee gut microbiota that are most targeted by prophages seem to be S. alvi and G. apicola. Also, in accordance to previous studies, the authors show that the variation in total number of prophages and prophage composition correlates with the host’s genome size. Finally, the authors identify several prophage-encoded genes that may supply their hosts with auxiliary benefits, such as genes in carbohydrate metabolism.

Overall, this manuscript is clear, straight-forward and professionally written. I have a few minor comments:

(1) Although the authors made good use of the publicly available data, I think the manuscript lacks a serious discussion of the limitations of this study. For instance, the low number of isolates for certain species (e.g. Firmicutes-4, and F. perrara) most probably limits the complete characterization of the prophage community. On the same line of reasoning, it looks that there might be a correlation between the number of bacterial isolates and the number of identified prophages for a given species. While the sample size is probably too low for any statistical analysis on the issue, I think the authors should address this point as it may represent a major bias to their analysis. In addition, the authors may discuss how mining metagenomics data for prophages may validate and complement their study.

(2) I am quite familiar with vConTACT2, nonetheless I struggled to understand Fig. 4A. In many cases, it is difficult to understand which nodes belong to the same subcluster and which nodes do not. For instance, are node 3 and 4 of the last row a subcluster or are they two separate subclusters? I think that adding edges between nodes of the same subcluster will solve this issue. Also, showing the position of the subclsters of Fig 4A in the network of Fig. 4B will help the reader to connect the two panels.

(3) In the discussion (line 426) the authors state, “Although honey bee gut bacterial communities contain relatively few bacterial phylotypes, prophages may contribute to the high bacterial strain diversity within the honey bee gut microbiome (Ellegaard et al., 2015; Ellegaard & Engel, 2019), as most prophage regions detected in this study were unique to a strain (Fig. 3)”.
However, I fail to understand how Fig. 3 supports this claim.

Experimental design

The authors adopted rigorous and conservative approach to the identification of prophages in the genomes of the 181 isolates. Most steps of their pipeline are well explained and supported by previous studies. However, I think that a clarification would benefit this manuscript:

(1) In my opinion, it is not clear which rationale stands behind the taxonomical classification of phages that were not already classified by vConTACT2. Starting from line 203, the authors state, “If a putative bee gut bacteria prophage’s top hit to a reference phage had an AAI > 70% across more than 50% of its proteins, it was considered a potential member of the family that the reference belonged to. If a prophage region shared 70% AAI across 20-50% of proteins with its top reference, it was classified as weakly related to its top reference’s family. Any predicted prophage regions that shared less than 70% AAI or less than 20% of all proteins with its top reference were considered unclassified.”
However, they provide no explanation in support of these thresholds. I understand that phage classification is a complex and constantly evolving topic. Nonetheless, the manuscript would benefit from a more rigorous explanation in support of the taxonomical classification. For instance, the authors may include a supplementary figure where they play with the threshold values to show how this affects their classification. Also, the authors may try to use other dedicated bioinformatics tools for taxonomic classification (for examples, see table 3 in https://journals.asm.org/doi/full/10.1128/mmbr.00004-21).

Validity of the findings

This study is an important and needed addition to the characterization of the honey bee gut virome. Indeed, previous studies on this topic focused on metagenomics data. This type of analysis is heavily biased in favor of lytic phages, since bacteria, together with their prophages, are filtered away before DNA extraction. Thus, to date there were no serious explorations of the honeybee gut microbiota prophage community. Bueren et al. take the first step in this direction by characterizing 303 prophages infecting 181isolates of honeybee gut-associated bacteria. Notably, they provide evidences that these prophages may offer auxiliary genes to their hosts, which may have significant impact on the honeybee gut microbiota. Therefore, this study highlights the need to take into account prophages to understand the contribution of phage-host interaction in the ecology of the honeybee gut microbiota.

Reviewer 3 ·

Basic reporting

Predicted prophages and genes , needs to be shared

Experimental design

Experimental design is very thorough

Validity of the findings

Clearly novelty and benefit to a range of researchers

Additional comments

The work is generally very well written and a pleasure to read for the most part. It details the presence of prophages in the bacterial genomes from the honeybee, and how these may contribute to the honeybee gut microbiome. The analysis of the number and diversity of prophages, is of interest.
The works suggests prophages may also contribute to auxiliary metabolic process of the host. While AMGs are clearly of interest, if these are really AMGs is not clear and how they may contribute to metabolism is not clear. The authors have explained in detail how the prophages were predicted from existing genomes. The data from this work is likely of great value to other researchers. However, currently without repeating the entire process the phage genomes and genes cannot be easily extracted. The predicted prophages and genes should be made available (eg Figshare ) and when genes are described as an AMG, be clearly identifiable. In addition it is not clear how many of these genes might alter host metabolism. Further explanation for some of them would aid understanding
Line 59 – the berg et al reference is not the first to suggest this. It originates from Hendrix , Hendrix RW. 2002. Bacteriophages: evolution of the majority. Theor Popul Biol 61:471–480. doi: 10.1006/tpbi.2002.1590.
L96 these families are no longer used. While citing previous research it would be useful to highlight these are no longer used. See ICTV guidance
L122 – why only dsDNA phages ? Inoviruses have been referenced (phage ctx) but are not dsDNA ..
L125 why 5 kb ? when smaller prophages are known to occur ? eg phi CTX, as discussed in the introduction. A justification would help here.
L184 -why such as small dataset to compare against other phages ? RefSeq (despite the name) is not at all representative of phage diversity with another 15,K complete phage genomes in genbank that can be compared against.
L196 – Now Caudoviricetes rather than Caudovirales
L257 – the description of some of these genes as “auxiliary” genes is some what confusing . The authors have used auxiliary gene and not auxiliary metabolic genes. If this is intentional or not, is unclear. But it does read as if they infer they have an AMG function. But as described many of these genes do not, TA systems. The work of Sullivan and Breitbart who first coined this term, and subsequent work of Sullivan et al, provides a clear description of what an AMG might be.
A Nif-like gene, might fit this , but not TA genes. There are methods such as v-DRAM to identify AMGs – while this is probably overly strict. It does appear several non AMGs have been described as AMGs when they are not. It is not possible to identify the genes that are described as AMGs, providing this make the use of the analysis of far more use to others.


L324 – Please update taxonomy to use latest terms. The Myovorividae/podoviriade/siphoviriade are no longer families (and haven’t been for 10 months) The ICTV page contains latest taxonomic terms.
Please use italics for all ICTV taxonomy, per ICTV guidance
L334- minor here and elsewhere. Spaces between numbers and units
L346 – Given the previous studies are based on predicted hosts. Could this not also be due to the large issues of predicting phage hosts. Thus, it is not known if they are rare or not ?
L348 – not clear what is meant by a predatory phage. Are there non-predatory phages ?
L357 – it would help to clarify what is meant by free phage. The viromes in the other studies could contain both “free” phages that are lytic or temperate (originating from prophages). Is it known which ? are these induced prophages ?

L383 – please provide the details of exactly which phage and which genes in supplementary information. A table with all predicted AMGs, in which prophage and a unique identifier, that can be found in a fasta/gbk file
L385 – please speculate how this might occur. Are these all in the same pathway? Are they likely limiting steps? The genes may well be beneficial, but there is no discussion of how or why.
L405 please update taxonomy to current taxonomy. These families were abolished in 2021 (Archives of Virology (2022) 167:2429–2440)
L408 the authors cite the work that describes the removal of the taxonomy they use. Which seems slightly odd, that they are aware that the taxonomy used is abolished. But still choose to use it. Given the latest ICTV taxonomy was released before this analysis was completed - dates of analysis are suggested as July 2022. Please use the current taxonomy https://ictv.global/taxonomy. As the statements are no longer true.

Figure 5. It is not clear how this kind of analysis is informative. 1) the Myoviriade, Podoviridae, Siphoviridae are no longer in use. Even before the large-scale changes to these families this year, the families Autographviridae, Ackermannviridae etc were accepted and in use. While I have some sympathy in the constant changes in taxonomy, many of these changes were ratified years prior to this analysis (Autographviridae, Ackermannviridae etc). These established families were not included in the analysis. 2) Surely the point of using taxonomy is to classify things into groups that are the same. Thus, things are either a Myoviridae or not . Having weakly does not seem to help with classification

---

## Round 0.2 · Minor Revisions

Thank-you for your revised submission. I will be happy to approve your manuscript for publication after two issues are addressed. First, you refer in your rebuttal letter to your data being placed in an archive at Virginia Tech, and provide a DOI for that data. But a statement indicating the availability of the data and the DOI linking to that data are not provided in the manuscript. Please add this information about data availability to the manuscript.

Second, I am concerned about the statement in your rebuttal letter that "...any hits to these now defunct families were manually reclassified as "unclassified". While the myo, podo, and sipho viridae families may have been abolished, all species previously classified under these families have been reclassified into other ICTV families, or into subfamilies or genera unassigned to a family. Therefore, none of the previously classified viruses would be unclassified. They have all been reclassified into a different higher rank.

---

## Round 0.3 · accepted · Accept

Thank-you for responding to the most recent critique of your manuscript and addressing all concerns. I am happy to accept your paper for publication in PeerJ.